# Preparation and Characterization of Degradable Cellulose−Based Paper with Superhydrophobic, Antibacterial, and Barrier Properties for Food Packaging

**DOI:** 10.3390/ijms231911158

**Published:** 2022-09-22

**Authors:** Xiaofan Jiang, Qiang Li, Xinting Li, Yao Meng, Zhe Ling, Zhe Ji, Fushan Chen

**Affiliations:** 1College of Marine Science and Bioengineering, Qingdao University of Science and Technology, Qingdao 266042, China; 2Jiangsu Co−Innovation Center of Efficient Processing and Utilization of Forest Resources, College of Chemical Engineering, Nanjing Forestry University, Nanjing 210037, China; 3State Key Laboratory of Biobased Material and Green Papermaking, Qilu University of Technology, Shandong Academy of Sciences, Jinan 250353, China

**Keywords:** food packaging, cellulose paper, polylactic acid (PLA), super−hydrophobicity, antibacterial

## Abstract

A great paradigm for foremost food packaging is to use renewable and biodegradable lignocellulose−based materials instead of plastic. Novel packages were successfully prepared from the cellulose paper by coating a mixture of polylactic acid (PLA) with cinnamaldehyde (CIN) as a barrier screen and nano silica−modified stearic acid (SA/SiO_2_) as a superhydrophobic layer. As comprehensively investigated by various tests, results showed that the as−prepared packages possessed excellent thermal stability attributed to inorganic SiO_2_ incorporation. The excellent film−forming characteristics of PLA improved the tensile strength of the manufactured papers (104.3 MPa) as compared to the original cellulose papers (70.50 MPa), enhanced by 47.94%. Benefiting from the rough nanostructure which was surface−modified by low surface energy SA, the contact angle of the composite papers attained 156.3°, owning superhydrophobic performance for various liquids. Moreover, the composite papers showed excellent gas, moisture, and oil bacteria barrier property as a result of the reinforcement by the functional coatings. The Cobb_300s_ and WVP of the composite papers were reduced by 100% and 88.56%, respectively, and their antibacterial efficiency was about 100%. As the novel composite papers have remarkable thermal stability, tensile strength, and barrier property, they can be exploited as a potential candidate for eco−friendly, renewable, and biodegradable cellulose paper−based composites for the substitute of petroleum−derived packages.

## 1. Introduction

Plastic pollution is a rising serious and global ecological issue. The widespread abuses in single−use consumer plastics that are used for food or drug packages have been a major cause for the concern [1]. Most disposable food packaging items are made of petroleum−based, non−degradable thermoplastics, which has given rise to increasing environmental contaminations and threats to human health. In 2015, the annual consumption of single−use plastic packages was over 407 million tons and kept increasing trend year by year [2]. Due to its recalcitrance to biodegradation, plastic packages not only cause white pollution and soil hardening but also flow into the ocean forming permanent microplastics [3]. Scientists have discovered microplastics in the placentas of human newborns and adult blood, which is the criminality for numerous deadly diseases [4]. Therefore, developing low ecotoxicity and biodegradable food packages to replace petrochemical based single−use plastics has become a research focus. It is imperative to find biodegradable alternatives to compostable plastic packaging [5].

In recent decades, enormous efforts have been devoted to exploiting biodegradable and high−quality food packages from cellulose, starch, polycaprolactone, and polybutylene adipate−co−terephthalate (PBAT) [6,7]. The cellulose paper−based materials manufactured from lignocellulose fibers particularly stand out due to the advantages of biosafety, biodegradability, vast availability, and low cost [8,9]. Food packaging is aimed at safeguarding food quality and preventing food from being damaged during transportation and sale. Except for the mechanical strength, the barrier to water and steam is one of the most significant properties. However, cellulose contains many hydrophilic hydroxyl groups. Additionally, the three−dimensional networks of fiber papers can jointly induce capillary action, resulting in a high hygroscopicity [10]. These intrinsic features limited the ability of papers to moisture barrier, thus deteriorating their mechanical performance as well as food shelf−life [11].

To overcome the defects, internal sizing, or surface treatment of papers with functional coatings, has been adopted. Considering the cost and feasibility, exploiting coated composite paper packages with natural polymers is a promising strategy [12]. Among the natural polymers, polylactic acid (PLA) has attracted much attention owing to its non−toxicity, biocompatibility, and excellent mechanical properties. The high crystallinity and excellent film−forming characteristics endow PLA with favorable barrier qualities [13,14]. Especially the presence of vast ester linkages in PLA grants its high resistance to water [15,16,17]. Rhim et al. manufactured a PLA−coated paperboard with improved tensile strength, water resistance, and heat sealing qualities, which could be a substitute for PE−coated paper [18]. Additionally, PLA can create a weakly acidic environment on the film surface to restrain bacterial and mold [19]. Moderate incorporation of natural antibacterial agents will further reinforce its activity [20,21,22,23,24]. The chemical compound cinnamaldehyde cinnamonaldehyde (CIN) is widely utilized in the production of feed, cosmetics, fragrances, and medications. With considerable bactericidal effects on Gram−positive bacteria (*E. coli*) and Gram−negative bacteria (*S. aureus*), broad−spectrum antibacterial capabilities, and significant efficacy on fungus, CIN has essential uses in bactericidal disinfection and antisepsis [25,26]. CIN is recognized as “Generally Recognized as Safe” (GRAS) by the Food and Drug Administration and current European legislation (EU N10/2011 Regulation), which means it can be used in food processing without further approval [27]. Villegas et al. proposed a supercritical impregnation procedure to introduce cinnamaldehyde into PLA sheets for improving its antibacterial property, while the method was high−cost [28].

However, single−component PLA is brittle and shows a limited hydrophobic property. For food packaging, super−hydrophobicity can expand its applications in the areas of re−humidity, barrier, and self−cleaning. Two critical strategies for achieving superhydrophobic characteristics are nanoscale rough textures and low surface energy compounds [29]. Chen et al. coated hydrophobic silica nanoparticles onto the surface of PLA films to achieve a contact angle of 150.2° for self−cleaning packaging [30]. Sbardella et al. used ZnO and stearic acid (SA) to treat PLA/linen fabric green composites to achieve high tensile strength and a superhydrophobic fiber surface of 150° [31]. However, few alterations that would give the superhydrophobic paper−based food packages a barrier to gas molecules and water vapor have been suggested in previous investigations. In order to extend the passage path of water molecules and improve the solid’s ability to block the passage of water vapor, balanced polymer structural morphology and tortuous paths are necessary [32,33].

In the present work, we developed an easy process for producing antibacterial, biodegradable, and superhydrophobic composite paper packages. The as−designed packages were prepared from the cellulose paper by coating a mixture of PLA/CIN as a barrier screen and nano−SiO_2_−modified SA as a superhydrophobic layer. The schematic illustration of the preparation is displayed in Figure 1. The chemical structure, thermal stability, mechanical property, morphology, superhydrophobic property, water vapor, oxygen, and oil barrier properties, as well as antibacterial efficiency, were comprehensively investigated by a series of advanced characterization techniques.

## 2. Results and Discussion

### 2.1. FT−IR Analysis

FT−IR analysis was conducted to investigate the chemical structures of composite papers. As illustrated in Figure 2a, the (CH_2_)_n_ of SA is represented by the absorption peak at 1186–1313 cm^−1^. The peaks at 2916 cm^−1^, 2846 cm^−1^, 1706 cm^−1^, and 1470 cm^−1^ indicate the stretching vibration of CH_3_, CH_2_, C=O, and the in−plane bending vibration of CH of SA, respectively [34]. The vibrations of Si−O−Si are responsible for the larger absorption bands between 850–1100 cm^−1^ [35]. The peak in the SA/SiO_2_ curve, which overlaps and extends with the curve of SA stretching vibration, is present at many of the above locations. The OH symmetric stretching and bending vibration at 3300–3500 cm^−1^ is attenuated in the spectrum of SA/SiO_2_, suggesting the interaction of SiO_2_ and SA (Figure 1b).

In Figure 2b, the specific bands at 3280 cm^−1^, 2890 cm^−1^, 1320 cm^−1,^ and 1020 cm^−1^ were characteristic peaks of cellulose papers [28]. After coating by PLA/CIN, the 1450 cm^−1^ and 1358 cm^−1^ peaks of CH_3_ symmetric bending and the 1743 cm^−1^ of the unique carbonyl group (C=O) are detected as the typical bands of PLA [36]. A distinctive peak of CIN can be collected at 1680 cm^−1^, which is induced by the stretching vibration of R−CHO [37]. In the curve of PC−SA, PLA was essentially covered due to the spraying modification of SA. It shows that the COOH absorption peak emerges faintly at 1706 cm^−1^. The peaks at 2890 cm^−1^ and 2851 cm^−1^ represent the vibration of CH_3_ and C−H in SA. In the spectra of PC−SA/SiO_2_, Si−O−Si bonds have vibration absorption maxima at 1100 cm^−1^ and 860 cm^−1^, respectively. It coincides with the curve of SA stretching vibration and broadens, demonstrating that SiO_2_ nanoparticles and SA work well together, which is also consistent with the trend of the curve in Figure 2a.

### 2.2. Thermal Stability and Mechanical Properties

The thermal stability of cellulose paper and coated paper was investigated by TGA and DTG analysis under a nitrogen atmosphere. According to TGA curves (Figure 3a), all samples show great pyrolysis−related similarities that were divided into three stages. First, when the temperature rises to about 170 °C and levels off, mass loss begins to happen for cellulose paper. For the other samples, there is an initial mass loss below 290 °C due to the decomposition of water. After that in the temperature range of 290–390 °C, a significant mass loss driven by the decomposition of cellulose paper and PLA can be observed. Finally, with rising temperature the curves essentially remain unchanged, resulting in consistent quality. Among them, the cellulose paper is completely decomposed at 547 °C, while the other samples retain some ash at the end of the TGA experiment. The constant mass of PC−SA/SiO_2_ is higher than that of the others because of the incorporated inorganic SiO_2_.

From the DTG curves (Figure 3b), the lower peak temperature of PC compared to CP is due to the addition of CIN to PLA, which has a plasticizing effect on the polymer base material and increases the fluidity of the polymer chains. It improved the flexibility and ductility of the material, which resulted in a decrease in thermal properties [13,37]. In contrast, the thermal degradation temperature of PC−SA/SiO_2_ increased accompanied by the peak temperature rising from 355 to 360 °C, virtually matching that of CP. It indicates that the presence of hydrophobic nanoparticles enhances the material’s crystallinity, thus promoting its thermal stability. The results suggest that the material’s outstanding thermal stability can be ensured by the superhydrophobic modified samples for use in food packaging.

The stress−strain curves of cellulose paper and coated paper are shown in Figure 3c. Benefiting from the PLA, PLA interactions on the surface of papers and its excellent film−forming characteristics, the TS of PC raised to 118.9 MPa increased by 68.65% as compared to that of CP (70.5 MPa). The decrease in breaking elongation might be due to physical interaction between PLA and CIN (Figure 3d), which restricts PLA chain mobility to some extent [38]. Unexpectedly, a slight reduction in the TS of superhydrophobic modified composites was observed. A reason for the phenomenon is that the unsatisfactory surface free energy of the sprayed particles at the interface of multilayers limited the interfacial adhesive strength between the multi−coatings. Thus, when the samples are subjected to external loads, the stress cannot be effectively transferred to the inorganic particles, but instead, the bearing cross−sectional area of the paper is reduced, lowering the tensile strength of the composite papers [39]. Even so, they do outperform the original paper in terms of mechanical strength with PC−SA and PC−SA/SiO_2_ rising by 43.83% and 47.94%, respectively (Appendix A). The results suggested the improved mechanical strength of the coated composite papers than the base one.

### 2.3. SEM and Superhydrophobic Properties

SEM and CA were used to examine the surface morphology and contact angle states of the original and coated paper samples. The paper was porous and the fiber texture can be obviously distinguished (Figure 4a). The capillary action and the exposed OH groups on the fiber surface are easier to absorb water with a lower CA of 85°. After coating PLA/CIN, the surface became smooth and compact, which indicated the excellent film−forming ability at the interface (Figure 4b). The CA of PC samples (90.1°) was slightly higher than the base papers (Figure 4c). For the superhydrophobic modified samples, rough surface structure can be observed (Figure 3d,e). SA has low surface energy, a multi−vacancy surface, and a micro−nano structure. The features help to increase the surface’s hydrophobicity [34]. With a contact angle of 130.9°, the surface of the coated paper spray with SA has more “clusters” gathered on it, making the surface structure rougher and increasing the area of air trapping, thereby enhancing hydrophobicity (Figure 4d). after multilayer coating, the generated paper surface showed more dense shape. The loading of nanoparticles enhances the contact area with air and decreases the surface energy, resulting in a greater hydrophobicity with a contact angle of 156.3° (Figure 4e).

Figure 5a shows that the water contact angle of the base cellulose paper is much lower (85°) than expected due to its intrinsic hydrophilic. Benefiting from a rough surface, the PC−SA/SiO_2_ possesses the best super−hydrophobicity with a contact angle of 156.3°. To measure the hydrophobicity of as−prepared packages to different liquids, the states of several liquids poured on the sample surfaces were captured (Figure 5b). Water, oolong tea, red sparkling water, and blue sparkling water are the four transparent liquids on the left, while coffee, milk, methyl blue stain, and methyl red stain are the four opaque liquids on the right, all of which have near−perfect sphericity. Figure 5c demonstrates how water sprayed from a straw onto a superhydrophobic surface bounces back instantly and leaves no trace, exhibiting excellent superhydrophobic qualities and a weak contact between water and the modified sample surface. In Movie M1, this is also proven. When utilized as the inside wall of containers for various liquid food products, the created superhydrophobic material can decrease waste by reducing residue.

The self−cleaning property of a superhydrophobic surface is an important attribute, as it swiftly eliminates dirty dust from the surface and keeps it clean. Carbon black powder was sprinkled over a superhydrophobic surface tilted at 20°, and then water was progressively dripped onto the contaminated surface to investigate the changed samples’ self−cleaning properties (Figure 5d). As can be seen, water droplets may readily carry dust along a path, providing a visible path delineated by dotted lines. A heavy layer of toner powder was applied to the droplets. Finally, the water droplets fully remove the carbon black powder and totally clean the surface, displaying the surface’s excellent self−cleaning capability. The self−cleaning effect was well demonstrated in Movie M2. This also demonstrates that the material may be used as a packing material and is dust and dirt−proof on the exterior surface, indicating that it has a wide range of applications.

All the above phenomena indicated that because the surface of the super−hydrophobically modified sample constitutes micro/nanoscale asperities with air trapped between it and water, it exhibited excellent superhydrophobic properties, which was in accordance with the Cassie−Baxter model.

### 2.4. Barrier Performance

To prevent food quality degradation, food packaging must have strong water, oil, and gas barrier qualities. Modification, as demonstrated in Figure 6 and Table 1. Figure 6a demonstrates that after coating with PLA, the cellulose paper has superior water resistance, which is owing to PLA’s intrinsic hydrophobicity [40]. The larger the contact angle with water after the superhydrophobic alteration, the better the waterproof performance of the samples, as indicated in Appendix A. When compared to the original paper, the Cobb value of sample PC−SA/SiO_2_ can be lowered by 100% in 60–300 s. Even when the test duration is prolonged to 600 s, the Cobb value is 1.2, which is 82.75% lower compared with CP. While extending the time to 1800 s, the cobb value of sample PC−SA/SiO_2_ can still be kept below 10, which is a 50.51% decrease compared to CP. The results show that the superhydrophobic modified samples have excellent water resistance.

The thickness and gram weight of the coated and modified cellulose papers coupled with their corresponding barrier properties were indicated in Table 1. The porous structure of the base cellulose paper led to poor oil resistance with a valley Kit number of 0, while the oil resistance of all modified samples was enhanced to the greatest level (Kit number 12). In order to evaluate the effectiveness of coatings on the gas barrier, the base, and coated papers were subjected to a 24 h oxygen transmission test. The oxygen transmission rate (OTR) of CP was out of the measuring range of the instrument (500,000 cm^3^/m^2^·24 h·0.1 MPa) due to its loosened structure as illustrated by SEM. As expected, the OTR of the coated and superhydrophobic modified papers was sharply dropped to a minimum value of 57.942 cm^3^/m^2^·24 h·0.1 MPa. The water vapor permeability (WVP) was another key parameter for assessing the barrier performance. The WVP of PC−SA/SiO_2_ (206.95 g/m^2^·24 h·Pa) was greatly reduced as compared to the original paper (1802.35 g/m^2^·24 h·Pa). CIN that is evenly diffused in PLA works as an efficient heterogeneous nucleating agent and stimulates the production of PLA crystals [41]. Coupled with the excellent film−forming characteristic of PLA and surface nanomodification, the transport pathway of gas and water vapor is changed and inhibited to some extent [42]. Compared with the commonly used non−degradable plastic food packaging materials in Table 2, PC−SA/SiO_2_ shows comparative oxygen barrier performance, although its WVP was not so satisfactory. While referring to the recently reported self−developed food packages, our novel composite paper package exhibits outstanding barrier performance to oil, gas, and water vapor.

### 2.5. Antibacterial Properties

The antimicrobial performance of packages is an important property that was usually measured by observing how well the samples inhibited the development of *E. coli* and *S. aureus*. The antibacterial activity of samples was assessed by comparing the colonies number in the test and control groups at a certain time. As shown in Figure 7, the inhibitory action of CIN−rich samples is obviously improved compared to the PLA−coated papers. The hydrophobic nature of CIN allows it to permeate the phospholipid bilayer of bacteria, disorganizing its cell walls and releasing internal contents [46]. Meanwhile, CIN can also stop cells from breathing by disrupting associated enzymes, resulting in cell death [47]. PC exhibits 91.94 ± 0.52% and 97.19 ± 0.92% inhibition against *E. coli* and *S. aureus*, respectively. The capacity to suppress *S. aureus* was superior to that of *E. coli*. A possible explanation is that the outer barrier surrounding the cell walls of *E. coli* prevents chemicals from accessing their lipopolysaccharide layers, rendering them resistant [37]. It should be noted that the samples that functioned with super−hydrophobicity had stronger antibacterial activity, which shows the higher hydrophobicity, the better antimicrobial capabilities. The reason is that super−hydrophobicity lowers the attachment of bacteria to solid surfaces, allowing germs to be readily removed before biofilm formation [48]. PC−SA inhibited *E. coli* and *S. aureus* growth by 99.18 ± 0.23% and 99.31 ± 0.46%, respectively, whereas the antibacterial efficiency of samples modified by PC−SA/SiO_2_ to both bacteria reached up to 100% (Appendix A).

## 3. Materials and Methods

### 3.1. Materials

Polylactic acid (PLA 4032D), in pellets, was obtained from Nature Works with Mw = 1.0 × 10^5^. The hydrophobic nano−silica (SiO_2_, 99%) was supplied by Macklin (Shanghai, China) with a specific area of 300 ± 50 m^2^/g and an average particle size of 15 nm. Stearic acid (SA, 95%) and cinnamaldehyde (CIN) were acquired from Aladdin. Dichloromethane (DCM) was purchased from the National Medicine Group Chemical Reagents Co., Ltd., Suzhou, China. All the chemicals were of analytical grade and were used without further purification. 

### 3.2. Sample Preparation

A solution of 10% PLA in DCM was stirred at room temperature for 1 h and then CIN and span80 were dissolved in the above solution with stirring for 10 min to prepare the antibacterial coating solutions. Subsequently, the antibacterial coating solutions were coated on cellulose paper’s surface using a coating rod model RDS−24 on the automatic film coating instrument (BEVS 1811) at a rate of 30 mm/s, and the solvent was evaporated under natural conditions. To prepare solution I, 1 g of stearic acid was dissolved in 100 mL of ethanol solution. Modification of cellulose paper is sprayed by solution I. 2 g hydrophobic nano−silica was added to Solution I and stirred for 2 h at room temperature to make Solution II. Solution II was sprayed on the coating paper to yield a superhydrophobic surface. During the spraying process, the synthesized superhydrophobic solution was uniformly coated on the surface of the paper samples using a spray gun with an air pressure of 30 psi from 30 cm [49]. The freshly coated surface was then dried at 60 °C for 5 min. The coating and drying process was repeated 3 more times to provide a homogeneous and thick coating. The samples are labeled as follows: CP is cellulose paper, PC is PLA/CIN coated paper, PC−SA is PLA/CIN coated paper with SA spraying, and PC−SA/SiO_2_ is PLA/CIN coated paper with SA/SiO_2_ spraying.

### 3.3. Characterization Techniques

FT−IR (Nicolet IS10, Waltham, MA, USA) was used to determine the functional groups of the samples, and the wavenumber range was set at 600–4000 cm^−1^. Thermal characterization of the materials was carried out by a TGA (TG 209 F3 Tarsus, Bavaria, German) from 30 to 600 °C at a heating rate of 10 °C/min under a nitrogen atmosphere. The tensile strength and elongation at the break of the samples were collected d by a tensile testing machine (AI−3000−UL, Taiwan, China). SEM (regulus 8100, Tokyo, Japan) was used to investigate the surface morphology of papers.

To value the hydrophobicity, a goniometer (JC2000D1, Shanghai, China) was utilized to ascertain the contact angle. The contact angle of the modified papers was observed using different transparent and opaque liquids attached to the sample surface. In the self−cleaning test, the carbon−black powder was deposited on the inclined (approximately 20°) sample surface with deionized water passing over its surface until all carbon dust was washed off. Water is shot onto the surface of the hydrophobic paper to observe hydrophobicity. The status of the modified hydrophobic paper was observed using different drops of transparent and opaque liquids. The grammage of films was calculated based on the ratio between their mass and area (g/m^2^).

### 3.4. Barrier Properties

The oil resistance of cellulose paper and coated papers was evaluated according to a TAPPI standard test T559 pm−96 using a Kit test procedure [50]. For this, a series of solutions with different Kit numbers (1–12) were created by specific proportions of castor oil, toluene, and *n*−heptane. The highest−numbered solution that remained on the surface of the coated papers without causing staining was reported as the Kit number for the coated papers. Kit number 1 represented the weakest oil resistance and Kit number 12 represented the strongest oil resistance. Water absorptiveness (WA) was tested by an ACT2500 automatic Cobb water absorption tester (BTG, Stockholm, Sweden). Water vapor permeability (WVP) was measured by W3/010 analyzer (Labthink, Shandong, China) following GB/T 16928 standard. The oxygen transmission rate (OTR) of the paper samples was determined according to GB/T 1038–2000 standard by VAC−V3 pressure difference gas permeameter (Labthink, Shandong, China).

### 3.5. Antibacterial Efficacy

The antimicrobial performance of the samples was evaluated using PLA−coated paper as a control according to “GB4789.2−2016“. All samples were cut into slices and sterilized under a UV lamp for 30 min. First, *E. coli* and *S. aureus* were incubated in the LB medium for 24 h to obtain a bacterial suspension at a concentration of 10^7^ CFU/mL. Then the bacterial suspension was diluted to 10^5^ CFU/mL with normal saline. Additionally, 0.1 mL of the test bacterial solution was added dropwise to the samples. Three parallels were made for each sample. After covering the samples with sterilized film to make the bacteria evenly touch the samples, we place them in sterilized dishes and incubate them at (37 ± 1) °C with a relative humidity greater than 90% for 24 h. Remove the samples from the 24 h incubation, add 20 mL of eluent, wash the samples and covering film repeatedly, shake well, and then inoculate a certain amount into nutrient agar medium at (37 ± 1) °C. The live bacteria were counted after 24–48 h incubation.

## 4. Conclusions

A novel kind of degradable composite paper package with super−hydrophobicity and excellent barrier properties was successfully fabricated. By coating a mixture of PLA/CIN as a barrier screen and nano−SiO_2_−modified SA as a superhydrophobic layer, the as−prepared packages possessed excellent thermal stability. Additionally, the tensile strength of the manufactured papers reached up to 104.3 MPa attributed to the excellent film−forming characteristics of PLA and the compatibilizer effect, and the PLA−SA interactions at the multilayer interface. As a result of the rough surface formed by SA/SiO_2_, the CAs of the composite papers attained 156.3°, showing superhydrophobic performance for various liquids and self−cleaning properties. The bifunctional coatings also changed the surface absorbability and transport pathway of small molecules, resulting in an excellent barrier performance for oil, gas, and water vapor. The antibacterial efficiency of the novel composite paper was about 100%, which suggested a promising prospect in the degradable packaging application.

## Figures and Tables

**Figure 1 ijms-23-11158-f001:**
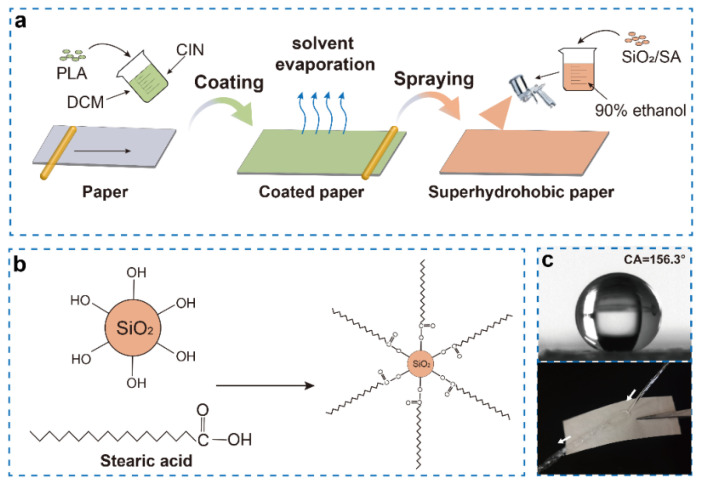
Schematic illustration of the preparation of degradable cellulose−based superhydrophobic antibacterial materials (**a**); mechanism diagram of nano−SiO_2_ modified stearic acid (**b**); contact angle and bounce of water column on the superhydrophobic paper (**c**).

**Figure 2 ijms-23-11158-f002:**
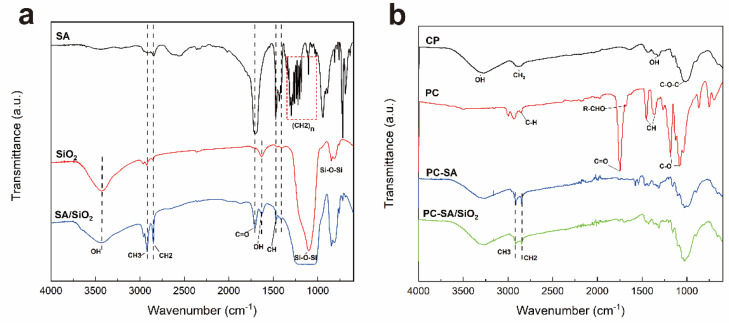
FT−IR spectra of SA (stearic acid), SiO_2_, SA/SiO_2_ (**a**) and CP (cellulose paper), PC (coated paper), PC−SA (coated paper with SA spraying), PC−SA/SiO_2_ (coated paper with SA/SiO_2_ spraying) (**b**).

**Figure 3 ijms-23-11158-f003:**
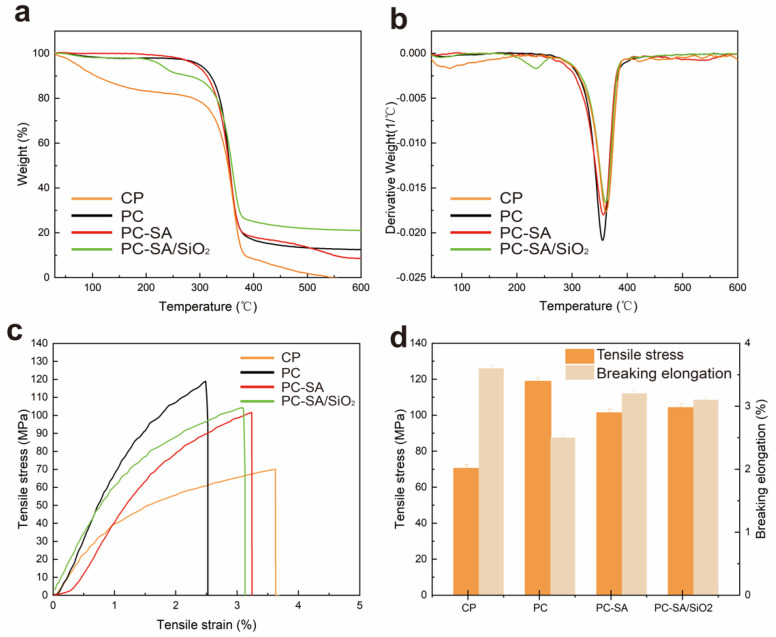
Thermogravimetric curves (**a**), differential thermogravimetric curves (**b**), tensile stress−strain curves (**c**), and breaking elongation (**d**) of samples.

**Figure 4 ijms-23-11158-f004:**
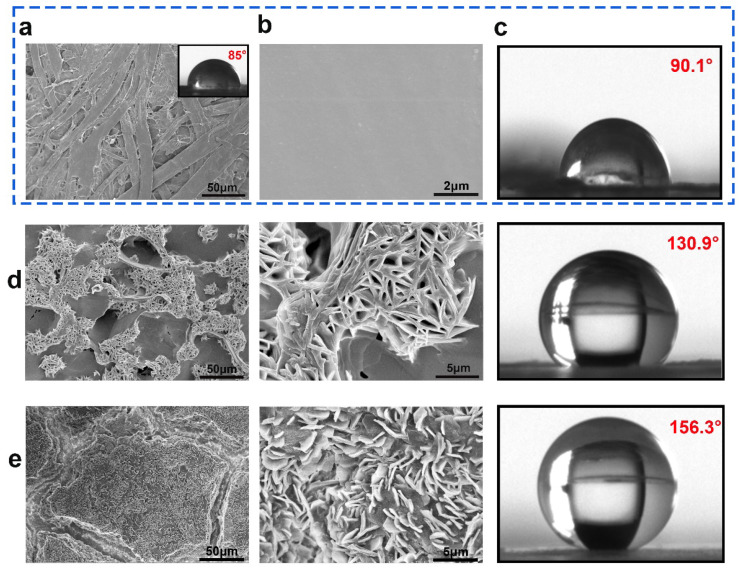
SEM and CA images of samples: CP (**a**); PC (**b**,**c**); PC−SA (**d**); PC−SA/SiO_2_ (**e**).

**Figure 5 ijms-23-11158-f005:**
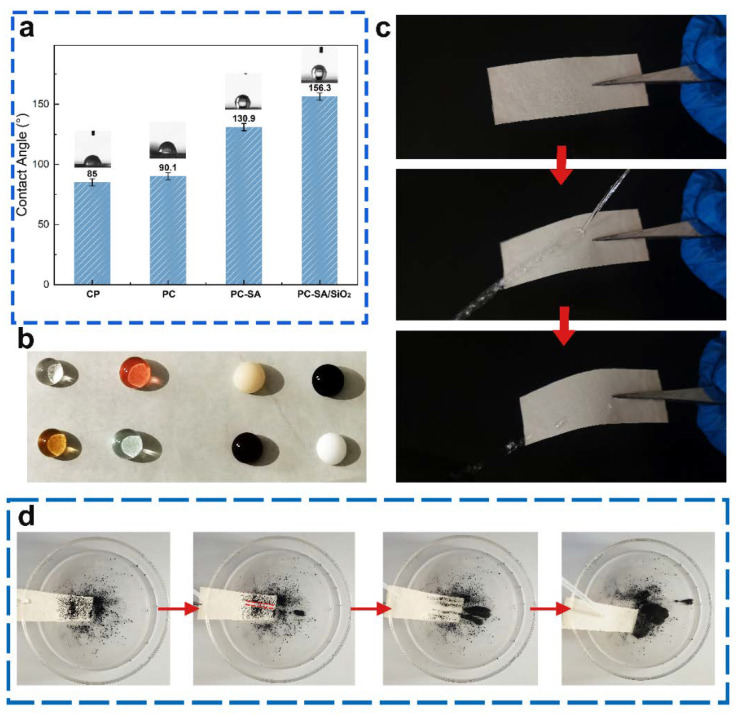
Photographs show the water contact angle of samples (**a**), state of different liquids (**b**), bouncing with the water column (**c**), and self−cleaning properties of PC−SA/SiO_2_ (**d**). The red arrows indicate the chronological order of the experiments.

**Figure 6 ijms-23-11158-f006:**
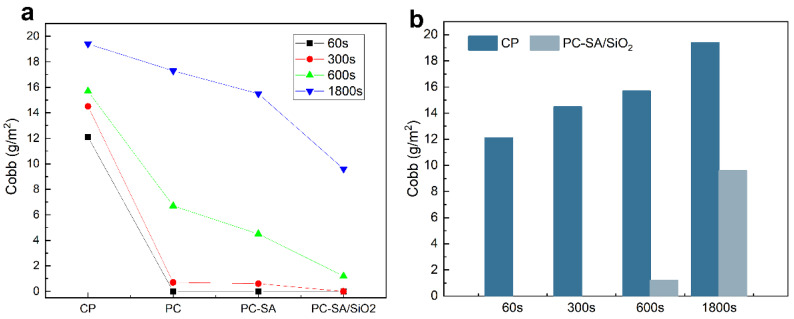
Waterproof performance curve of CP, PC, PC−SA and PC−SA/SiO_2_ with test time (**a**), and comparison of CP and PC−SA/SiO_2_ in water resistance (**b**).

**Figure 7 ijms-23-11158-f007:**
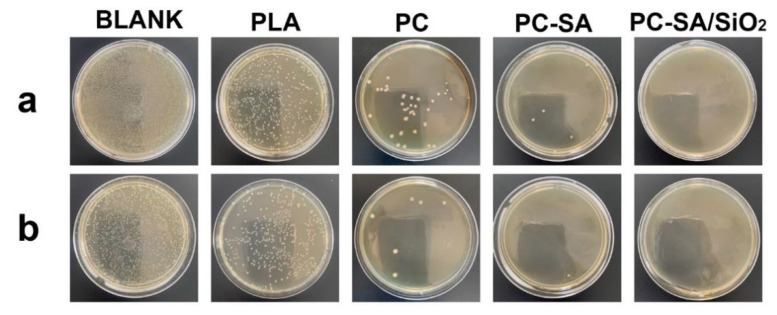
Antibacterial response of different samples after 24 h of incubation against *E. coli* (**a**) and *S. aureus* (**b**).

**Table 1 ijms-23-11158-t001:** Thickness, gram weight, and barrier properties of cellulose paper and coated paper.

Sample	Thickness (μm)	Grammage (g/m^2^)	Kit	OTR (cm^3^/m^2^·24 h·0.1 MPa)	WVP (g/m^2^·24 h·Pa)
CP	32.7	30.64	0	/	1802.35
PC	47.3	42.32	12	143.237	315.21
PC−SA	52.6	45.45	12	79.315	265.11
PC−SA/SiO_2_	54.6	45.49	12	57.942	206.95

**Table 2 ijms-23-11158-t002:** Comparison of barrier performance of PC−SA/SiO_2_ with the commonly used food packaging materials and recently reported self−developed food packages.

Sample	Other Substances	Thickness (μm)	Grammage (g/m^2^)	OTR (cm^3^/m^2^·24 h·0.1 MPa)	WVP (g/m^2^·24 h·Pa)	References
LDPE	/	20–80	/	98–453	25	[43]
HDPE	/	20–80	/	26.3–98.5	5–62.5	[43]
PTFE	/	50	/	222–387	0.09–6	[43]
PP	/	20–60	/	35–377	6.7–10	[43]
PGA/PBAT	GMA	95.4–123	/	59.8–74.3	236.9–277.5	[43]
Nanofiber film	PLA	/	/	/	1029.36	[17]
PLA	CIN	/	/	3120	27.59	[23]
Kraft paper	Acetylated cellulose	284	90.2	2600	20.84	[27]
Cellulose	Naringin	70	308.8	55	3000	[44]
PLA	CMCS/CMC	505.8	309.2	924	166	[45]
PLA/ZnONPs	CMCS/CMC	505.8	/	406	72	[45]
PC−SA/SiO_2_	/	54.6	45.49	57.942	206.95	This work

## Data Availability

Not applicable.

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
