# Peer review of "Preparation and Characterization of Degradable Cellulose−Based Paper with Superhydrophobic, Antibacterial, and Barrier Properties for Food Packaging"

_ijms, 2022, doi:10.3390/ijms231911158_

Round 1
Reviewer 1 Report
I liked the work presented by the authors. After careful evaluation, I suggest minor revisions before it can be accepted for publication. The comments I am suggesting will enhance the current presentation and clarity of the draft.
1. The major concern in plastics pollution is the increased use of single-use plastics. Please add this in the first paragraph to highlight the necessity of compostable and biodegradable alternatives. Authors should refer to DOI: 10.1021/acssuschemeng.1c01424. The last line of the first paragraph has to be rewritten. Instead of eco-friendly, the author should use 'low eco-toxic and biodegradable packaging to replace petrochemical based single-use plastics'. This is important because eco-friendly packages can be derived from natural sources and still lead to plastic pollution. (e.g., bioplastics like bioPP, bioPET, etc.) The authors should refer to DOI: 10.1039/D0GC01394C
2. The barrier performance of their materials can be compared with other barrier materials for food packaging, not just PLA, kraft paper, and cellulose-based systems. Please refer to DOI: 10.1021/acssuschemeng.1c07376 and compare it with your system. Put the barrier table in S3 into the main text.
3. Please mention the biocidal efficiency of PC with S.D. because the samples would have been tested in triplicates. Figure 7 is not a zone of inhibition test. From the materials and methods section, it appears that the authors performed a standard plate count of their system. I cross-checked with the standard mentioned. Please change figure description 7 to ' Antibacterial response of different samples after 24 h of incubation against E.coli (a) and S.aureus (b)'.
4. Please check the bacteria concentrations 10^7 is written as 107, and 10^5 is written as 105 CFU/mL.
Author Response
请参阅附件。

Reviewer 2 Report
The present manuscript demonstrates a surface modification of cellulose papers using SiO2 and PLA and stearic acid to improve their properties. The SiO2 and PLA provide texture and mechanical strength, while stearic acid ensures hydrophobicity. The materials were characterized by TGA, FTIR and SEM while various other properties such as tensile strength, water contact angle, water vapor and oxygen barrier properties of the modified papers demonstrated enhanced performance of the PC-SA/SiO2 modified paper. The finding presented in the manuscript is interesting and well written. The manuscript should be considered for publication after a few minor revisions.
Some comments are.
1. The authors have used dichloromethane as a solvent which is toxic to the environment and biological systems, therefore calling the process completely green is not true. Did the authors try any other solvent i.e., methanol?
2. In SEM images, image 4a seems unusually large. Are the authors sure, they have used the same magnification image?
3. For Carolina Villegas et al. and others, use Last name and et al.
4. The role of cinnamaldehyde is not clear. Does it solely used for its antibacterial properties? Cinnamaldehyde can also polymerize to a hydrophobic long chain. How much is the contribution of cinnamaldehyde in the hydrophobicity?
